# Sepsis and the Human Microbiome. Just Another Kind of Organ Failure? A Review

**DOI:** 10.3390/jcm10214831

**Published:** 2021-10-21

**Authors:** Kevin M. Tourelle, Sebastien Boutin, Markus A. Weigand, Felix C. F. Schmitt

**Affiliations:** 1Department of Anesthesiology, Heidelberg University Hospital, 420, Im Neuenheimer Feld, 69120 Heidelberg, Germany; kevin.tourelle@med.uni-heidelberg.de (K.M.T.); markus.weigand@med.uni-heidelberg.de (M.A.W.); 2Department of Infectious Disease, Medical Microbiology and Hygiene, University Hospital, 324, Im Neuenheimer Feld, 69120 Heidelberg, Germany; sebastien.boutin@med.uni-heidelberg.de

**Keywords:** microbiome, microbiota, sepsis, ARDS, postoperative complications, next-generation sequencing, 16S rRNA gene sequencing

## Abstract

Next-generation sequencing (NGS) has been further optimised during the last years and has given us new insights into the human microbiome. The 16S rDNA sequencing, especially, is a cheap, fast, and reliable method that can reveal significantly more microorganisms compared to culture-based diagnostics. It might be a useful method for patients suffering from severe sepsis and at risk of organ failure because early detection and differentiation between healthy and harmful microorganisms are essential for effective therapy. In particular, the gut and lung microbiome in critically ill patients have been probed by NGS. For this review, an iterative approach was used. Current data suggest that an altered microbiome with a decreased alpha-diversity compared to healthy individuals could negatively influence the individual patient’s outcome. In the future, NGS may not only contribute to the diagnosis of complications. Patients at risk could also be identified before surgery or even during their stay in an intensive care unit. Unfortunately, there is still a lack of knowledge to make precise statements about what constitutes a healthy microbiome, which patients exactly have an increased perioperative risk, and what could be a possible therapy to strengthen the microbiome. This work is an iterative review that presents the current state of knowledge in this field.

## 1. Introduction

In the 1970s, Sanger, Maxam, and Gilbert developed the technique of DNA sequencing. Since then, the process has been further optimised. Today, next-generation sequencing (NGS) enables a higher capacity in sequencing, with faster results and lower costs than the first molecular methods.

In 2010, a study established a catalogue of 3.3 million non-redundant human intestinal microbial genes [1]. NGS is a novel procedure among molecular methods for pathogen diagnostics, and it is able to analyse the overall DNA fragments in the sample. The procedure can also differentiate between human DNA, bacteria, eukaryotes, archaea, and chloroplasts. In 2013, Skvarc et al. investigated molecular methods for pathogen diagnostics using PCR. At that time, the conclusion was that, despite certain limitations, molecular methods should be further investigated together with culture-based methods in everyday clinical practice [2]. Since the NGS method has broader applications than previous molecular methods, this statement is more valid than ever. NGS was also able to determine which bacteria were present in the blood of critically ill patients within 30 h. The results coincided with the results of blood cultures, and NGS also showed other microbiota that were not detectable in blood cultures [3]. In comparison, conventional blood cultures take between 24 and 120 h from collection to results [4]. Cell-free DNA found in NGS may also result from the translocation of DNA by gut bacteria into the blood. Therefore, NGS results from blood samples should be interpreted critically. Initial work on cell-free DNA sequencing with optimised workflows and nanopore-based real-time sequencing, which further develops NGS, detected bacteraemia within 6 h. It was also shown that this method achieved a detection rate of 90.6% in 239 retrospectively evaluated sepsis samples by extrapolation. Furthermore, by using NGS, the study investigated a six-fold increased positive rate for pathogen hits in the course of sepsis compared to blood cultures [5]. In another study with septic patients, NGS results would have resulted in a change in antibiotic therapy in 53% of cases [6]. Moreover, in pneumonia patients in an ICU, NGS could detect microbiota in 84% of the patients, whereas culture-based methods could only detect microbiota in 65% of the patients [7]. Otto et al. were able to show that there are three phases during sepsis and that these are probably related to changing pathophysiological mechanisms. In the late phase of sepsis, there is an increase in positive blood cultures [8]. In combination with the previously presented results, showing that blood cultures only select some of the microbiota, NGS methods could provide earlier and further insights into the pathophysiological mechanisms of sepsis and a possible organ failure. However, it must be emphasised that NGS results must then be examined to determine whether the pathogens found were viable or dead pathogen residues. For this purpose, the wet lab or bioinformatics must reconcile the results critically.

This review aims to summarise new insights in perioperative microbiome analysis, its impact on the individual course of the critically ill patient in the perioperative setting, the possible benefits in clinical use, and which gaps research still has to investigate. In more detail, we present the current clinical situation for sepsis, ARDS, and general complications which may lead to sepsis as well as the extent to which the microbiome could improve diagnostics and reasons why a possible therapy to strengthen the microbiome could be beneficial like in other organ failures. In addition, the first therapeutic approaches are also presented and discussed.

## 2. Materials and Methods

In March 2021, a literature search was performed using the well-established PubMed database to create an informative review. Studies were included in which perioperative complications were investigated. Since septic and critically ill patients and the study of the microbiome is a field of research that has only emerged in recent years, an iterative approach was used to select suitable papers. Animal models, case series, controlled and uncontrolled studies, and meta-analyses were included. Case reports were also included to provide an overview of the perioperative situation. The focus was placed on the period from 2014. The keywords used were “microbiome”, “microbiota”, “perioperative complications”, “sepsis”, “pneumonia”, “ARDS” or “16S RNA”. Only English-language papers were cited for the review evaluation. In the first step, suitable papers were searched for using the keywords above. Subsequently, the abstract and title were screened and, if the papers were suitable, the full text was examined. Studies were excluded if they had nothing to do with sepsis, perioperative complications, or evidence of an unhealthy or healthy microbiome. You can also find an overview about the included papers in Table 1.

## 3. Results

### 3.1. Sepsis-Induced Acute Respiratory Distress Syndrome

A study published in 2016 found that 10.4% among all patients admitted to the ICU and 23.4% of all ICU-ventilated patients suffered from an ARDS [30]. The current mortality rate of ARDS remains high and is up to 43% [31], even though there is still no consensus on what is meant by a healthy lung microbiome. Specific microorganisms are often found in patients with healthy lungs [9]. However, the detection of these microorganisms seems to be challenging in clinical practice. Even in critically ill ARDS patients who required mechanical ventilation, almost 43% of bronchoalveolar lavage cultures were negative, although NGS showed a positive result [10]. A lower alpha diversity indicates poor lung health, and the alpha diversity also decreases by the time of invasive ventilation [10,11,12,13]. Ventilated ARDS patients showed a significantly lower alpha diversity and a higher dominance with only one bacterial species (>50%) when their BAL cultures were also positive compared to BAL-negative patients [14]. Kyo et al. demonstrated that ARDS patients had a decreased alpha diversity of the lung microbiome with increased in-hospital mortality. Additionally, they demonstrated that ARDS had abundant Betaproteobacteria, *Staphylococcus*, *Streptococcus*, and Enterobacteriaceae, which may play a pivotal role in the pathophysiology of ARDS patients [13]. Patients who had nursing-home and hospital-associated infection (NHAI) pneumonia had significantly more reduced alpha diversity than non-NHAI pneumonia patients [15]. Cigarette smoking trauma patients are more likely to suffer from an ARDS. They have a different microbiome with a higher abundance of *Streptococcus*, *Fusobacterium*, *Prevotella*, *Haemophilus*, and *Treponema* compared to non-smokers [16]. Smoking significantly alters the lung microbiome. In a mouse model, it was shown that mice exposed to smoke for 2 h each day for 90 days had increased levels of *Staphylococcus*, *Acinetobacter*, and *Bacillus*, all are considered pathogenic microorganisms. [17]. Finally, it is essential to distinguish whether a microorganism is responsible for a disease or just a part of the colonisation. For example, in pneumonia patients who had a positive BAL culture for methicillin-resistant *Staphylococcus aureus* (MRSA), it could be shown that this is often only colonisation without disease value [18], which makes the interpretation of microbiome data in a clinical setting a lot more demanding.

In the pathogenesis of sepsis and ARDS, a gut-lung axis has often been described. In a murine model, typical gastrointestinal bacteria were detected in a septic ARDS in the pulmonary tract, such as Bacteroidales order, *Enterococcus* species, and Lachnospiraceae species [19]. Whether this translocation is part of an unrecognised bloodstream infection or is the result of ongoing gastric reflux followed by microaspirations remains unclear.

### 3.2. Perioperative Sepsis and Complications

Anastomotic insufficiencies with resulting peritonitis are a common reason for a severe sepsis and can lead to organ failure. Changes in the microbiome composition seem to be associated with the incidence of these insufficiencies. Patients who underwent bowel surgery and showed a reduced alpha diversity were more prone to higher infection rates and anastomotic complications [20]. Patients with colorectal anastomoses showed reduced alpha diversity and the abundance of Lachnospiraceae and Bacteroidacae [21,22]. Furthermore, a microbial imbalance with a higher dominance of several bacteria also increased the risk of anastomotic leakage. [32].

In 32 patients who underwent pancreatic surgery, three distinct microbiome enterotypes were found. One of the enterotypes showed an increase in Enterobacteriaceae, *Akkermansia*, and *Bacteroidales* and a decrease in *Bacteroides*, *Prevotella*, and Lachnospiraceae. Patients that have shown colonisation with this specific enterotype at least once during their hospital stay suffered significantly more often from complications and had a significantly longer hospitalisation. [23].

A study conducted on liver transplant patients also demonstrated the advantage of NGS in diagnosing fungal infections. The invasive fungal disease could be distinguished from colonisation by comparing NGS and culture-based results [24]. Furthermore, it was possible to distinguish a fungal infection from colonisation in critically ill sepsis patients using NGS, thus demonstrating the usefulness of NGS to detect invasive fungal disease [25].

### 3.3. Therapeutic Influences on the Microbiome

Typical medications in critically ill sepsis patients are antibiotics, sedatives, and opioids. However, opioids also appear to affect the microbiome. Patients who received opioids and no antibiotics had a detectable dysbiosis of the gut microbiome. A control group in patients who did not receive opioids had increased protective *Blautia* and *Lactobacillus*. Moreover, piperacillin/tazobactam appears to have a particularly negative effect on the gut microbiome. Patients with this antibiotic regimen seem to have a lower abundance of protective bacteria such as *Lactobacillus* and *Clostridiales* [26].

In a review by van Ruissen et al., original papers were screened and distinguished between patients who received a probiotic diet to change their microbiome compared to patients who did not receive such a diet and determined the percentage that developed ventilator-associated pneumonia during their stay in the ICU. Due to the heterogeneity of the underlying papers, no accurate statistical analysis could be performed. However, a trend showed that 13–50% of patients who did not receive a specific probiotic diet developed ventilator-associated pneumonia. In contrast, 4–36% of critically ill patients who received a probiotic diet developed VAP [33].

A mouse model showed that a high-fibre rodent diet could ensure that the inflammatory response to sepsis was lower than in mice without a special diet [27]. A pilot study showed no significant differences in beneficial short-chain fatty acids in critically ill patients on broad-spectrum antibiotic therapy who received a regular or high-fibre diet. There was just a trend towards a high-fibre diet and increased beneficial short-chain fatty acids, but the number of cases was small (10 patients per group) [28]. This reflects that altering the gut microbiota can have an impact on sepsis and its complications. An individualised diet adapted to the patients, their microbiome, their disease, their pre-existing conditions, and their general condition should be the goal of avoiding the loss of alpha diversity and an accompanying dysbiosis [34]. Treatment with probiotics containing *L. plantarum* reduces infections in critically ill patients [35]. A combination of probiotics and prebiotics, which include non-digestible fibre and can promote the growth of certain bacteria, are called synbiotics. The administration of synbiotics in critically ill patients increased beneficial bacteria such as Bifidobacterium and Lactobacillus in the stool. At the same time, there were significantly fewer cases of enteritis and VAP in the group receiving synbiotics. However, an advantage in terms of mortality could not be shown [29].

An overview of the most important key findings of perioperative sepsis, sepsis-induced ARDS, and the influence of drugs and probiotics is shown in Figure 1.

## 4. Discussion

An increasing number of studies have shown that the microbiome is altered in disease. Thus, it can also be assumed that septic patients suffer from an altered microbiome. The causes of sepsis are numerous and can lead to organ failure. So far, typical triggers are perioperative complications, such as VAP, anastomotic insufficiencies, catheter infections, organ failure and ischemia. Not to be neglected is also the patient’s initial situation, such as underlying diseases, medication, and nutritional status. In this context, the microbiome in the perioperative setting could also be a factor for patient outcomes in the future. Furthermore, the preoperative microbiome may be used for risk assessment, and changes in the postoperative phase may be a sign of patients who are at risk. The question arises of whether we have to focus more on treating imbalances in the patient’s microbiome, like in other organ failures.

Tuddenham and Sears explain that there is a bi-directional relationship between the host and the microbiome. A healthy microbiome is made up of many factors. These factors not only include diet, age, and host genetics as well as antibiotic use and its long-acting effects on the microbiome, but also microbiome composition and diversity, greater bacterial richness, as well as resistance to stressors and resilience, i.e., the ability to return to equilibrium after a stressor has upset the microbiome. It should not be underestimated that a healthy and stable microbiome can positively influence both the immune system and metabolism [36].

The idea that the lungs are sterile has existed for a long time, but today, we know that the lung microbiota exists for various microorganisms. Many of them seem to be translocated from the oropharyngeal tract [37]. The microbiome of the lungs is shaped from birth. In the first years of life, the microbiome’s composition can already cause a predisposition to later diseases such as asthma, lung infections, and allergies. In general, the lung microbiome of healthy people differs from that of people with diseases. In addition, the lungs are exposed to the environment and can be influenced by allergens, microorganisms, and pollutants. Furthermore, the gut microbiome can affect the balance of the lung microbiome through the gut−lung axis by the immune system [38]. Therefore, preserved alpha diversity appears to play a key role in critically ill patients’ outcomes because an increased alpha diversity reflects a healthy microbiome [13]. In addition, with or without sepsis, critically ill patients seem to have a markedly altered microbiome compared to healthy patients. It is also not uncommon that the microbiome in these patients consists of 50% and sometimes more than 75% of only one bacterial genus [39], which paves the way for problematic bacteria such as *Pseudomonas aeruginosa*. Given the previous results, it seems that the lung microbiome is, to a certain extent, related to the oral microbiome. However, at the same time, this relationship must be balanced to prevent serious infections. An increased presence in the lungs of *Haemophilus influenza, Moraxella catarrhalis*, and *Streptococcus pneumoniae*, especially at a young age, leads more frequently to asthma. These results are further evidence of an existing gut−lung axis and its interaction [40].

That the microbiome and various bacteria interact with the immune system was shown in an in vitro study with macrophages and *Chlamydia pneumoniae*. After 24 h, increased *C. pneumoniae* DNA was detected in M2 macrophages compared to M1 macrophages so that *C. pneumoniae* could not be eliminated by pro-inflammatory M1 macrophages. Thus, anti-inflammatory M2 macrophages seem to form a niche for *C. pneumoniae*. The possible persistence of *C. pneumonia* in M2 macrophages is another indication of the interdependence of the microbiome and our immune system and that dysbiosis or infection with a pathogen can contribute to a long-acting disruption of our immune system [41].

Unfortunately, no microbiome baseline with specific characteristics has yet been found in a healthy person’s microbiome. However, some basic assumptions are now being discussed. These include, for example, the Firmicutes/Bacteroidetes ratio, which showed in a study of intensive care patients that an F/B ratio of <0.1 or >10 had a poorer outcome and that none of the survivors demonstrated such a value [42]. However, there are studies that have not demonstrated this effect [23,39]. In addition, there appears to be continuous transport from the upper to the lower airway. Thus, there is a balance between immigration, colonisation, and clearance of oral bacteria entering the lungs physiologically. Hence, the abundance of *Veillonella* spp. and *Prevotella* spp. was frequently detected in healthy individuals [37]. Furthermore, certain risks could be identified, e.g., an increased risk for Candida infection during immunosuppression if there was a Candida colonisation in the mouth before [43,44]. Pulmonary colonisation with Candida species is an independent risk factor for VAP with *A. baumanii* [45]. Moreover, Candida colonisation in ICU patients resulted in pulmonary dysbiosis during the course [46]. A microbiome baseline would be desirable to identify patients at risk. Not only pathogenic or high-risk microbiota could be identified, but the knowledge of protective microbiota could also help patients in the perioperative setting.

Recent studies show that other drugs besides antibiotics can also have an impact on the microbiome. For example, proton pump inhibitors affect the incidence of *Clostridioides difficile* infections and an increased risk of community-acquired pneumoniae with *Streptococcus pneumoniae* and other enteric infections. Metformin causes adverse gastrointestinal effects (such as diarrhoea) in 30% of patients suspected to be caused by metformin-induced dysbiosis. However, antipsychotics also have side effects such as increased visceral fat and weight gain, triggered by increased interleukin-8 and interleukin-1ß. These are at least the primary results in the murine model investigating the microbiome in association with antipsychotics [47]. During sepsis, critically ill patients often receive a wide range of medications. These include anti-infectives, catecholamines, sedatives, opioids, anticoagulants, and others. Whether and to what extent these drugs have an impact on the microbiome and consequently on the outcome during sepsis remains unclear. It also needs to be clarified whether, for example, the reduced gut motility due to opioids and not the opioids themselves are responsible for a lowered alpha diversity.

The individual’s lifestyle and comorbidities also appear to be relevant factors in the composition of an individual microbiome. For example, age influences the composition of the microbiome during pneumonia; thus, elderly patients suffering from nursing-home- or hospital-acquired infection also appear to have reduced alpha diversity [15]. Patients older than 75 years have a significantly higher number of *Streptococcus* phylotypes than those aged 74 years and younger [48].

Oral gut dysbiosis is associated with metabolic and degenerative diseases [49]. These include metabolic syndrome, type 2 diabetes, and chronic neurodegenerative diseases [50,51]. In addition, intestinal dysbiosis has been shown in osteopenic and osteoporotic patients, with increased Firmicutes phyla and reduced Bacteroidetes [52]. This is explained, among other reasons, by changes in oxidative reactions and glycolysis [53,54]. A healthy microbiome maintains the natural barrier of the epithelium with its mucus shield, whereas dysbiosis, by destroying the barrier, can lead to an increased permeability and the migration of endotoxins. Dysbiosis is associated with neurodegenerative disorders such as Alzheimer’s and Parkinsons disease [55,56]. Furthermore, a dysbiosis−immune system/central nervous system (CNS) crosstalk is discussed. The vagus nerve between the CNS and the enteric nervous system is seen as a neuroendocrine link [51,56]. It also appears that it is necessary to study the entire microbiota and not only the bacterial microorganisms. Several studies have already shown that fungal and viral infections are associated with an increased abundance of bacteria [57,58]. In patients infected with influenza, an increased number of Pseudomonadales (not *Pseudomonas aeruginosa*) could also be detected in the upper respiratory tract. Further studies must show whether these have a pathological value. In the course of a viral infection and with its healing, the altered microbiome changes again in the direction of healthy individuals [58]. The extent to which viruses, fungi, and bacteria interact in critically ill patients and whether microorganisms are found to have a pathophysiological effect or are part of physiological colonisation must be shown in further investigations that examine, among other things, the molecular interactions between the microorganisms themselves.

Furthermore, the examination of the microbiome alone may not be sufficient. A healthy intestinal microbiome seems to protect against *Clostridioides difficile* infection, but there may also be other factors. For example, a metabolome study showed that the secondary bile acid deoxycholate acted as an inhibitor for *Clostridioides difficile*. Thus, secondary bile acid also appears to be a factor in *Clostridioides difficile* infection [59]. This study is another indicator that our microbiome interacts directly with our body, and the goal must be to keep this interaction in balance.

To keep the gut microbiome healthy, McClave et al. recommend “the delivery of early enteral nutrition, the provision of soluble fibre, and the generation of short-chain fatty acids”. As mentioned above, therapy with probiotics is discussed to achieve a healthy microbiome. It has been shown in vitro that *Lactobacillus plantarum*, *Lactobacillus rhamnosus GG*, and *Lactobacillus casei* have an anti-inflammatory effect on the epithelial tissue [51]. However, a large multicentre RCT could not demonstrate any impact between administering a probiotic with *Lactobacillus rhamnosus GG* compared to placebo on the incidence of ventilator-associated pneumonia and other infections such as bacteremia [60]. Therefore, it needs to be further investigated whether sepsis patients can also benefit from probiotics. Alternatively, faecal microbiome transplants should be considered for critically ill patients [61]. It was possible to successfully treat sepsis with diarrhoea and multi-organ failure in critically ill patients by faecal microbiota transplant (FMT) [62,63,64]. Simultaneously, there are also reports of drug-resistant *E. coli* infection transmitted through a faecal microbiome transplant and death in due course [65]. Therefore, FMT should not be undertaken lightly, and donors must be very carefully selected and screened. It may be possible to identify the composition of a healthy microbiome to initiate targeted therapies in the future.

Many studies have focused on the microbiome in sepsis or sepsis-related complications consisting of either animal models, case series, or relatively small, controlled studies. Therefore, application to clinical practice is still difficult. Multicentre studies with greater power need to further verify the previous results and provide more detailed insights into the microbiome of the septic patient.

## 5. Conclusions

In summary, severely ill patients, especially those with sepsis, suffer from an altered microbiome. Furthermore, the microbiome is very heterogeneous, and individual changes vary in septic patients, which makes it even more demanding to treat this kind of organ failure. Nonetheless, microbiome sequencing via NGS is a promising diagnostic method in the perioperative setting to screen patients preoperatively and identify those at risk for complications. NGS can distinguish bacterial, fungal, and viral infections from colonisations in critically ill patients. In addition, higher sensitivity and rapid availability of results compared to classical culture-based diagnostics should be emphasised here.

## Figures and Tables

**Figure 1 jcm-10-04831-f001:**
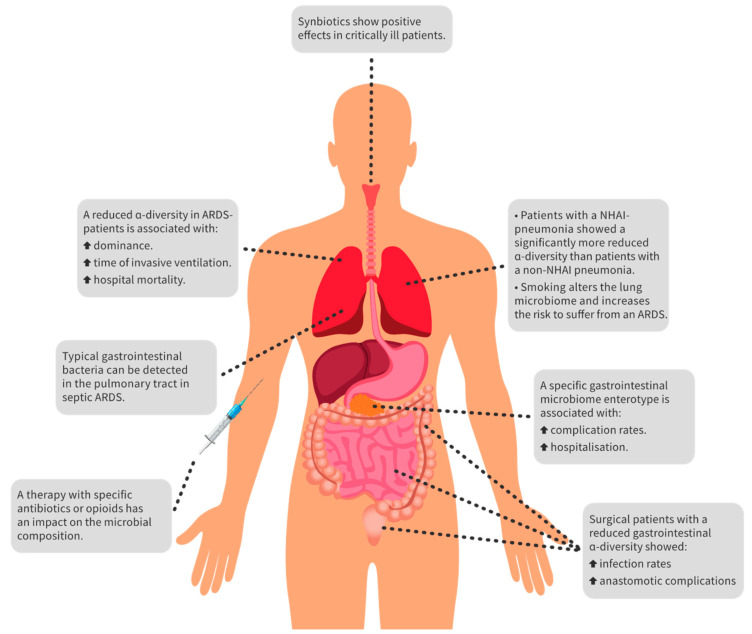
Overview of the key findings. There are many ways to influence the microbiome in critically ill patients with sepsis and ARDS. For example, antibiotics and opioids harm the balance of the microbiome. On the other hand, it appears that probiotics have a positive influence on dysbiosis. Furthermore, typical gastrointestinal bacteria can be detected in the lungs of ARDS patients and are indicative of lung dysbiosis. Overall, reduced alpha diversity in the gut and lungs is associated with more complications and increased hospital mortality. Abbreviations: ARDS, acute respiratory distress syndrome; NHAI-pneumonia, nursing-home and hospital-associated infection-pneumonia.

**Table 1 jcm-10-04831-t001:** Study characteristics.

Author	Year	Study Characteristics	Population	Sample Type	Methods	Outcome
Smith et al. [9]	2016	Uncontrolled trial	Human	Bronchoalveolar lavage (BAL)	15 mechanically ventilated patients in the surgical ICU	Streptococcus, Hydrogenophaga, and Haemophilus are among the most common genera Proteobacteria, Bacteroidetes, Firmicutes are dominant in culture-negative BAL
Schmitt et al. [10]	2020	Controlled trial	Human	BAL	15 ARDS patients vs. 15 lung-healthy patients	ARDS patients had significant lower alpha diversity False-negative cultures
Kelly et al. [11]	2016	Controlled trial	Human	Oropharyngeal (OP) and deep endotracheal (ET) secretions	15 intubated patients with respiratory failure Compared to a healthy control group	Ureaplasma parvum and Enterococcus daecalis were some dominant taxa that were not detected by culture Lower alpha diversity correlated with lower respiratory tract infection
Zakharkina et al. [12]	2017	Controlled trial	Human	Endotracheal aspirates (ETA)	35 intubated patients (11 with vs. 18 without ventilator-associated pneumonia)	Mechanical ventilation had an impact on the microbiome VAP patients had a stronger dysbiosis
Kyo et al. [13]	2019	Controlled trial	Human	BAL	47 intubated patients (40 with vs. 7 without ARDS)	ARDS patients had a significantly decreased alpha diversity Hospital mortality and serum IL-6 was decreased abundant Staphylococcus, Enterobacteriaceae, and Streptococcus
Kitsios et al. [14]	2018	Controlled	Human	ETA and plasma	56 patients 12 culture-positive vs. 44 culture-negative cases	In 20% negative culture-based diagnostics, abundant pathogens were found Decreased diversity is associated with >50% relative abundance of one taxon
Baek et al. [15]	2020	Controlled trial	Human	ETA	60 mechanically ventilated ICU patients 36 non-nursing-home- and hospital-associated infections (non-NHAI) vs. 24 nursing-home- and hospital-associated infections (NHAI)	Pneumonia had a significantly higher relative abundance of Corynebacterium Lower alpha diversity in NHAI-group
Panzer et al. [16]	2018	Uncontrolled trial	Human	ETA	74 mechanically ventilated patients after severe blunt trauma	Higher abundant Streptococcus, Prevotella, Treponema, Haemophilus, and Fusobacterium were associated with smoking Enterobacteriaceae, Fusobacterium and Prevotella were associated with the development of ARDS
Li et al. [17]	2019	RCT	Mice	Blood and lung tissue samples	20 male mice in smoking and 20 male mice in the non-smoking group	Increased pathogenic bacteria such as Bacillus, Acinetobacter, and Staphylococcus in smoking mice Proteobacteria phyla or Firmicutes phyla were associated with inflammation A negative correlation was found with IL-6 and CRP and less abundant Desulfuromonadales, Oceanospirillales, Lactobacillaceae, and Nesterenkonia
Kawanami et al. [18]	2016	Controlled trial	Human	BAL	42 patients with culture-positive MRSA 13 patients treated with anti-MRSA agents vs. 29 patients treated without anti-MRSA agents	No Staphylococcus aureus could be detected in 20 of 42 patients using BALF 28 of 29 patients treated without an anti-MRSA agent had a favourable course—possibly no MRSA pneumonia. The microflora analysis showed that in 19 of 28 patients, the Staphylococcus aureus phylotype had a minor part
Dickson et al. [19]	2016	Mice: RCT Human: uncontrolled trial	Mice and human	Mice: tissue, blood, and biomass specimens Human: BAL	68 patients with ARDS 10 only with antibiotic-treated mice vs. 10 mice treated with antibiotics and surgery after caecal ligation and puncture	During sepsis, the gut microbiome seems to be the starting point of pathogenic lung bacteria Bacteroides are gut-specific bacteria detected in the lungs during human ARDS TNF-alpha correlated significantly with an altered lung microbiome
Ralls et al. [20]	2014	Controlled trial	Human	Specimen of small bowel segment	15 samples (enterally fed vs. enterally deprived portions of the intestine vs. newborns)	Great variability of microbial diversity in all groups Systemic antibiotic therapy and loss of enteral nutrient intake is associated with decreased alpha diversity Low alpha diversity is associated with postoperative intestinal infections and anastomotic insufficiencies
van Praagh et al. [21]	2016	Controlled trial	Human	Colon and rectum tissue	8 patients with anastomotic leakage vs. 8 patients without anastomotic leakage	Lachnospiraceae was detected more frequently in patients with AL At the same time, body mass index correlated with the frequency of Lachnospiraceae Lower diversity in patients with AL
van Praagh et al. [22]	2019	RCT	Human	Colon and rectum tissue	60 patients received a C-seal anastomosis vs. 58 without a C-seal anastomosis	When patients did not receive a C-seal anastomosis, anastomotic leakage was associated with lower alpha diversity and abundant Bacteroidaceae and Lachnospiracea In the C-seal group, no difference in the microbiome was found in the development of anastomotic leakage
Schmitt et al. [23]	2019	Controlled trial	Human	Stool samples	32 patients undergoing pancreatic surgery 17 patients with vs. 15 patients without complications	Patients who had abundant Akkermansia, Bacteroidales, and Enterobacteriaceae and a decrease in Prevotella, Lachnospiraceae, and Bacteroides had a significantly higher risk of suffering postoperative complications No difference in alpha diversity between patients with and without complications
Decker et al. [24]	2019	Controlled trial	Human	Blood samples (plasma)	93 patients after liver transplantation 23 patients with fungal isolates vs. 70 patients without fungal isolates	NGS suitable for early identification of fungal pathogens through plasma samples in patients after liver transplantation Fungal SIQ score is a suitable tool to distinguish invasive fungal infections from colonisation
Decker et al. [25]	2017	Controlled trial	Human	Blood samples (plasma) (additionally: culture-based diagnostic procedures in ETA, drainage fluids, and wound swabs)	50 patients with sepsis 11 with fungal infection vs. 39 without fungal infection	NGS is suitable for diagnosing fungemia
Pettigrew et al. [26]	2019	Controlled trial	Human	Perirectal swabs	109 intensive care patients 41 patients with carbapenem-resistant Pseudomonas aeruginosa (CRPA) colonisation vs. 45 patients without CRPA colonisation vs. 23 patients without CRPA colonisation and without an antibiotic therapy	Patients treated with piperacillin−tazobactam had an increased risk of enterococcal dominance and had fewer protective bacteria such as B. Lactobacillus and Clostridiales and thus is particularly harmful Opioids were associated with dysbiosis Potentially protective bacteria such as Blautia were increased in patients not receiving opioids
Zhang et al. [27]	2018	RCT	Mice	Intestinal tissue	24 mice 8 diet with commercial normal-fibre rodent diet comprising normal fibre mice vs. 8 commercial normal-fibre rodent diet and underwent caecal ligation puncture (CLP) mice vs. 8 commercial high-fibre rodent diet mice vs. 8 commercial high fibre + CLP mice	A high-fibre diet improved survival after CLP A high-fibre diet lowered serum concentrations of pro-inflammatory cytokines Akkermansia and Lachnospiraceae were abundant when fed a high-fibre diet
Freedberg et al. [28]	2020	RCT	Human	Rectal swabs and stool samples	20 intensive care patients receiving broad-spectrum antibiotics 10 patients received 14.3 g/L fibre enteral nutrition vs. 10 patients received 0 g/L fibre enteral nutrition	A nonsignificant trend toward an increased abundance of bacteria producing short-chain fatty acids in patients on an enteral fibre diet
Shimizu et al. [29]	2018	RCT	Human	Rectal swabs	72 mechanically ventilated sepsis patients 35 patients received synbiotics vs. 37 patients received no synbiotics	Patients receiving synbiotics had significantly lower enteritis and ventilator-associated pneumonia incidence No difference in the incidence of bacteraemia Increased Bifidobacterium and Lactobacillus in the synbiotics group

## Data Availability

Not applicable.

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
