# Peer review of "Sepsis and the Human Microbiome. Just Another Kind of Organ Failure? A Review"

_jcm, 2021, doi:10.3390/jcm10214831_

Round 1

Reviewer 1 Report

Can the authors explain their strategy for literature review - most authors would do a wider search like Cochrane strategy

This is a potential weakness of the study

Suggestions

line 202 - the question arises

line 205 - the idea that the lungs are sterile

247 - we may be detecting....

268 - may also be

280 -undertaken lightly

286 - therefore application to clinical practice

291 - especially those with sepsis

293 - varied

297 - in addition,  higher sensitivity and rapid availability of results....

Author Response

Dear Reviewer,

Please find enclosed the revised version of the manuscript entitled “Sepsis and the human microbiome. Just another kind of organ failure? A review.” and a detailed point-to-point response to the reviewers’ comments.

We hope you will find the revised version appropriate for publication in Journal of Clinical Medicine.

Yours sincerely,

Felix Schmitt, MD

Response to the suggestions of Reviewer 1:

  1. Reviewer’s suggestion:

Can the authors explain their strategy for literature review - most authors would do a wider search like Cochrane strategy.

Authors’ statement:

An iterative approach was used to generate an overview about the current state of knowledge. To provide a better overview and to show as transparently as possible which papers were included, we inserted table 1. But indeed, we didn´t used a Cochrane strategy and this is a potential weakness of the study.

  1. Reviewer’s suggestion:

Suggestions:

  • line 202 - the question arises
  • line 205 - the idea that the lungs are sterile
  • 247 - we may be detecting....
  • 268 - may also be
  • 280 -undertaken lightly
  • 286 - therefore application to clinical practice
  • 291 - especially those with sepsis
  • 293 - varied
  • 297 - in addition,  higher sensitivity and rapid availability of results....

Authors’ statement:

We are very grateful for the nuanced input to this review and have incorporated the suggestions into our disscussion and conclusions as indicated below.

Related text section:

4. Discussion

An increasing number of studies have shown that the microbiome is altered in disease. Thus, it can also be assumed that septic patients suffer from an altered microbiome. The causes of sepsis are numerous and can lead to organ failure. So far, typical triggers are perioperative complications, such as VAP, anastomotic insufficiencies, catheter infections, organ failure and ischemia. Not to be neglected is also the patient's initial situation, such as underlying diseases, medication, and nutritional status. In this context, the microbiome in the perioperative setting could also be a factor for patient outcomes in the future. Furthermore, the preoperative microbiome may be used for risk assessment, and changes in the postoperative phase may be a sign of patients who are at risk. The question arises, if we have to focus more on treating imbalances in the patient's microbiome like in other organ failures.

Tuddenham and Sears describe that there is a bi-directional relationship between the host and the microbiome. A healthy microbiome is made up of many factors. These factors include diet, age, host genetics, antibiotic use and its long-acting effects on the microbiome, but also microbiome composition and diversity, greater bacterial richness, as well as resistance to stressors and resilience, the ability to return to equilibrium after a stressor has upset the microbiome. It should not be underestimated that a healthy and stable microbiome can positively influence both the immune system and metabolism [36].

The idea that the lungs are sterile has existed for a long time. But today, we know that the lung microbiota exists of various microorganisms. Many of them seem to be translocated from the oropharyngeal tract [37]. The microbiome of the lungs is shaped from birth. In the first years of life, the microbiome's composition can already predispose to later diseases such as asthma, lung infections and allergies. In general, the lung microbiome of healthy people differs from that of people with diseases. In addition, the lungs are exposed to the environment and can be influenced by allergens, microorganisms and pollutants. Also, the gut microbiome can affect the balance of the lung microbiome through the gut-lung axis by the immune system [38]. Therefore, preserved alpha diversity appears to play a key role in critically ill patients' outcome because an increased alpha diversity reflects a healthy microbiome [13]. Also, with or without sepsis, critically ill patients seem to have a markedly altered microbiome compared to healthy patients. And it is not uncommon that the microbiome in these patients consists of 50% and sometimes more than 75% of only one bacterial genus [39], which paves the way for problematic bacteria like Pseudomonas aeruginosa. Given the previous results, it seems that the lung microbiome is, to a certain extent, related to the oral microbiome. However, at the same time, this relationship must be balanced to prevent serious infections. An increased presence in the lungs of Haemophilus influenzaMoraxella catarrhalis, and Streptococcus pneumoniae, especially at a young age, leads more frequently to asthma. These results are further evidence of an existing gut-lung axis and its interaction [40].

That the microbiome and various bacteria interact with the immune system was shown in an in vitro study with macrophages and Chlamydia pneumoniae. After 24 hours, increased C. pneumoniae DNA was detected in M2 macrophages compared to M1 macrophages so that C. pneumoniaecould not be eliminated by pro-inflammatory M1 macrophages. Thus, anti-inflammatory M2 macrophages seem to form a niche for C. pneumoniae. The possible persistence of C. pneumonia in M2 macrophages is another indication of the interdependence of the microbiome and our immune system and that dysbiosis or infection with a pathogen can contribute to a long-acting disruption of our immune system [41]. 

Unfortunately, no microbiome baseline with specific characteristics has yet been found in a healthy person's microbiome. However, some basic assumptions are now being discussed. These include, for example, the Firmicutes/Bacteroidetes ratio, which showed in a study of intensive care patients that an F/B ratio <0.1 or >10 had a poorer outcome and that none of the survivors demonstrated such a value [42]. However, there are studies that have not demonstrated this effect [23,39]. In addition, there appears to be continuous transport from the upper to the lower airway. Thus, there is a balance between immigration, colonization, and clearance of oral bacteria entering the lungs physiologically. Hence, the abundance of Veillonella spp. and Prevotella spp. was frequently detected in healthy individuals [37]. Furthermore, certain risks could be identified, e.g. an increased risk for Candida infection during immunosuppression if there was a Candida colonisation in the mouth before [43,44]. Pulmonary colonisation with Candida species is an independent risk factor for VAP with A. baumanii [45]. Also, Candida colonisation in ICU patients resulted in pulmonary dysbiosis during the course [46]. A microbiome baseline would be desirable to identify patients at risk. Not only pathogenic or high-risk microbiota could be identified, but the knowledge of protective microbiota could also help patients in the perioperative setting.

Recent studies show that other drugs besides antibiotics can also have an impact on the microbiome. For example, proton pump inhibitors affect the incidence of Clostridiodes difficile infections and an increased risk of community-acquired pneumoniae with Streptococcus pneumoniae and other enteric infections. Metformin causes adverse gastrointestinal effects (such as diarrhoea) in 30% of patients suspected to be caused by metformin-induced dysbiosis. However, antipsychotics also have side effects such as increased visceral fat and weight gain, triggered by increased interleukin-8 and interleukin-1ß. These are at least the primary results in the murine model investigating the microbiome in association with antipsychotics [47]. During sepsis, critically ill patients often receive a wide range of medications. These include anti-infectives, catecholamines, sedatives, opioids, anticoagulants, and others. Whether and to what extent these drugs have an impact on the microbiome and consequently on the outcome during sepsis remains unclear. Or if we may detecting side-effects of the specific therapy, e.g. a decreased alpha-diversity due to reduced bowel motility caused by the use of opioids.

The individual lifestyle and comorbidities also appear to be relevant factors in the composition of the individual microbiome. For example, age influences the composition of the microbiome during pneumonia; thus, elderly patients suffering from nursing home- or hospital-acquired infection also appear to have reduced alpha diversity [15]. Patients older than 75 years have a significantly higher number of Streptococcus phylotypes than those aged 74 years and younger [48].

Oral gut dysbiosis is associated with metabolic and degenerative diseases. These include metabolic syndrome and chronic degenerative inflammation such as neurodegenerative and cardiovascular or type 2 diabetes. In addition, intestinal dysbiosis has been shown in osteopenic and osteoporotic patients, with increased Firmicutes phyla and reduced Bacteroidetes. This is explained, among other things, because the microbiome influences oxidative reactions and glycolysis. A healthy microbiome maintains the natural barriers of the endothelium with its mucus shield. Whereas dysbiosis, by destroying the barrier, can lead to increased permeability and the passage of endotoxins. Increased numbers of Coliform bacteria and Ralstonia genus combined with dysbiosis are associated with multiple sclerosis, Parkinson's disease, amyotrophic lateral sclerosis, and Alzheimer's disease. Furthermore, a dysbiosis-immune system/central nervous system (CNS) crosstalk is discussed. The vagus nerve between the CNS and the enteric nervous system is seen as a neuroendocrine link [49]. It also appears that it is necessary to study the entire microbiota and not only the bacterial microorganisms. Several studies have already shown that fungal and viral infections are associated with the increased abundance of bacteria [50,51]. In patients infected with influenza, an increased number of Pseudomonadales (not Pseudomonas aeruginosa) could also be detected in the upper respiratory tract. Further studies must show whether these have a pathological value. In the course of a viral infection and with its healing, the altered microbiome changes again in the direction of healthy individuals [51]. The extent to which viruses, fungi, and bacteria interact in critically ill patients and whether microorganisms found have a pathophysiological effect or are part of physiological colonisation must be shown in further investigations that examine, among other things, the molecular interactions between the microorganisms themselves.

Also, the examination of the microbiome alone may not be sufficient. A healthy intestinal microbiome seems to protect against Clostridioides difficileinfection, but there may also be other factors. For example, a metabolome study showed that the secondary bile acid deoxycholate acted as an inhibitor for Clostridioides difficile. Thus, secondary bile acid also appears to be a factor in Clostridioides difficile infection [52]. This study is another indicator that our microbiome interacts directly with our body and the goal must be to keep this interaction in balance.

To keep the gut microbiome healthy, McClave et al. recommend "the delivery of early enteral nutrition, the provision of soluble fibre, and the generation of short-chain fatty acids". As mentioned above, therapy with probiotics is discussed to achieve a healthy microbiome. It has been shown in vitro that Lactobacillus plantarum, Lactobacillus rhamnosus GG, and Lactobacillus casei have an anti-inflammatory effect on the epithelial tissue [49]. However, a large multicentre RCT could not demonstrate any impact between administering a probiotic with Lactobacillus rhamnosus GGcompared to placebo on the incidence of ventilator-associated pneumonia and other infections such as bacteremia [53]. Therefore, it needs to be further investigated whether sepsis patients can also benefit from probiotics. Alternatively, faecal microbiome transplants should be considered for critically ill patients [54]. It was possible to successfully treat sepsis with diarrhoea and multi-organ failure in critically ill patients by faecal microbiota transplant (FMT) [55-57]. Simultaneously, there are also reports of drug-resistant E. coli infection transmitted through a faecal microbiome transplant and death in due course [58]. Therefore, FMT should not be undertaken lightly, and donors must be very carefully selected and screened. It may be possible to identify the composition of a healthy microbiome to initiate targeted therapies in the future.

Many studies have focused on the microbiome in sepsis or sepsis-related complications consisting of either animal models, case series, or relatively small controlled studies. Therefore application to clinical practice is still difficult. Multicentre studies with greater power need to further verify the previous results and provide more detailed insights into the microbiome of the septic patient.

5. Conclusions

In summary, severely ill patients especially those with sepsis, suffer from an altered microbiome. Furthermore, the microbiome is very heterogeneous, and individual changes vary in septic patients, which makes it even more demanding to treat this kind of organ failure. Nonetheless, microbiome sequencing via NGS is a promising diagnostic method in the perioperative setting to screen patients preoperatively and identify those at risk for complications. NGS can distinguish bacterial, fungal, and viral infections from colonisations in critically ill patients. In addition, higher sensitivity and rapid availability of results compared to classical culture-based diagnostics should be emphasised here.

Reviewer 2 Report

This  is an interesting brief review about the potential role of our microbiota in sepsis development and evolution.

It is well focused and organized, but some other informations could be interesting for the readers. Please cite and discuss the following:

Tuddenham et al., PMID: 26237547 

Santacroce et al, PMID: 33019595

Isacco et al., PMID: 32727337

Buchacher et al., PMID: 26606059

At last, please revise the test for language and grammar style, and to correct certain typing errors as, but not only, the following:

Don't use italics for "spp.", and use it only for microbial names (i.e., Candida)

Lines 237-238 (and subsequent); C, difficile is currently named Clostridioides difficile

Line 267:  Don't use italics for "et al."

and more

Author Response

Dear Reviewer,

Please find enclosed the revised version of the manuscript entitled “Sepsis and the human microbiome. Just another kind of organ failure? A review.” and a detailed point-to-point response to the reviewers’ comments.

We hope you will find the revised version appropriate for publication in Journal of Clinical Medicine.

Yours sincerely,

Felix Schmitt, MD

Response to the suggestions of Reviewer 2:

  1. Reviewer’s suggestion:

It is well focused and organized, but some other informations could be interesting for the readers. Please cite and discuss the following:

Tuddenham et al., PMID: 26237547 

Santacroce et al, PMID: 33019595

Isacco et al., PMID: 32727337

Buchacher et al., PMID: 26606059

Authors’ statement:

We are very grateful for the constructive and helpful recommendations. We have incorporated the suggested papers into the discussion section as shown below.

Related text section:

4. Discussion

An increasing number of studies have shown that the microbiome is altered in disease. Thus, it can also be assumed that septic patients suffer from an altered microbiome. The causes of sepsis are numerous and can lead to organ failure. So far, typical triggers are perioperative complications, such as VAP, anastomotic insufficiencies, catheter infections, organ failure and ischemia. Not to be neglected is also the patient's initial situation, such as underlying diseases, medication, and nutritional status. In this context, the microbiome in the perioperative setting could also be a factor for patient outcomes in the future. Furthermore, the preoperative microbiome may be used for risk assessment, and changes in the postoperative phase may be a sign of patients who are at risk. The question arises, if we have to focus more on treating imbalances in the patient's microbiome like in other organ failures.

Tuddenham and Sears describe that there is a bi-directional relationship between the host and the microbiome. A healthy microbiome is made up of many factors. These factors include diet, age, host genetics, antibiotic use and its long-acting effects on the microbiome, but also microbiome composition and diversity, greater bacterial richness, as well as resistance to stressors and resilience, the ability to return to equilibrium after a stressor has upset the microbiome. It should not be underestimated that a healthy and stable microbiome can positively influence both the immune system and metabolism [36].

The idea that the lungs are sterile has existed for a long time. But today, we know that the lung microbiota exists of various microorganisms. Many of them seem to be translocated from the oropharyngeal tract [37]. The microbiome of the lungs is shaped from birth. In the first years of life, the microbiome's composition can already predispose to later diseases such as asthma, lung infections and allergies. In general, the lung microbiome of healthy people differs from that of people with diseases. In addition, the lungs are exposed to the environment and can be influenced by allergens, microorganisms and pollutants. Also, the gut microbiome can affect the balance of the lung microbiome through the gut-lung axis by the immune system [38]. Therefore, preserved alpha diversity appears to play a key role in critically ill patients' outcome because an increased alpha diversity reflects a healthy microbiome [13]. Also, with or without sepsis, critically ill patients seem to have a markedly altered microbiome compared to healthy patients. And it is not uncommon that the microbiome in these patients consists of 50% and sometimes more than 75% of only one bacterial genus [39], which paves the way for problematic bacteria like Pseudomonas aeruginosa. Given the previous results, it seems that the lung microbiome is, to a certain extent, related to the oral microbiome. However, at the same time, this relationship must be balanced to prevent serious infections. An increased presence in the lungs of Haemophilus influenzaMoraxella catarrhalis, and Streptococcus pneumoniae, especially at a young age, leads more frequently to asthma. These results are further evidence of an existing gut-lung axis and its interaction [40].

That the microbiome and various bacteria interact with the immune system was shown in an in vitro study with macrophages and Chlamydia pneumoniae. After 24 hours, increased C. pneumoniae DNA was detected in M2 macrophages compared to M1 macrophages so that C. pneumoniaecould not be eliminated by pro-inflammatory M1 macrophages. Thus, anti-inflammatory M2 macrophages seem to form a niche for C. pneumoniae. The possible persistence of C. pneumonia in M2 macrophages is another indication of the interdependence of the microbiome and our immune system and that dysbiosis or infection with a pathogen can contribute to a long-acting disruption of our immune system [41]. 

Unfortunately, no microbiome baseline with specific characteristics has yet been found in a healthy person's microbiome. However, some basic assumptions are now being discussed. These include, for example, the Firmicutes/Bacteroidetes ratio, which showed in a study of intensive care patients that an F/B ratio <0.1 or >10 had a poorer outcome and that none of the survivors demonstrated such a value [42]. However, there are studies that have not demonstrated this effect [23,39]. In addition, there appears to be continuous transport from the upper to the lower airway. Thus, there is a balance between immigration, colonization, and clearance of oral bacteria entering the lungs physiologically. Hence, the abundance of Veillonella spp. and Prevotella spp. was frequently detected in healthy individuals [37]. Furthermore, certain risks could be identified, e.g. an increased risk for Candida infection during immunosuppression if there was a Candida colonisation in the mouth before [43,44]. Pulmonary colonisation with Candida species is an independent risk factor for VAP with A. baumanii [45]. Also, Candida colonisation in ICU patients resulted in pulmonary dysbiosis during the course [46]. A microbiome baseline would be desirable to identify patients at risk. Not only pathogenic or high-risk microbiota could be identified, but the knowledge of protective microbiota could also help patients in the perioperative setting.

Recent studies show that other drugs besides antibiotics can also have an impact on the microbiome. For example, proton pump inhibitors affect the incidence of Clostridiodes difficile infections and an increased risk of community-acquired pneumoniae with Streptococcus pneumoniae and other enteric infections. Metformin causes adverse gastrointestinal effects (such as diarrhoea) in 30% of patients suspected to be caused by metformin-induced dysbiosis. However, antipsychotics also have side effects such as increased visceral fat and weight gain, triggered by increased interleukin-8 and interleukin-1ß. These are at least the primary results in the murine model investigating the microbiome in association with antipsychotics [47]. During sepsis, critically ill patients often receive a wide range of medications. These include anti-infectives, catecholamines, sedatives, opioids, anticoagulants, and others. Whether and to what extent these drugs have an impact on the microbiome and consequently on the outcome during sepsis remains unclear. Or if we may detecting side-effects of the specific therapy, e.g. a decreased alpha-diversity due to reduced bowel motility caused by the use of opioids.

The individual lifestyle and comorbidities also appear to be relevant factors in the composition of the individual microbiome. For example, age influences the composition of the microbiome during pneumonia; thus, elderly patients suffering from nursing home- or hospital-acquired infection also appear to have reduced alpha diversity [15]. Patients older than 75 years have a significantly higher number of Streptococcus phylotypes than those aged 74 years and younger [48].

Oral gut dysbiosis is associated with metabolic and degenerative diseases. These include metabolic syndrome and chronic degenerative inflammation such as neurodegenerative and cardiovascular or type 2 diabetes. In addition, intestinal dysbiosis has been shown in osteopenic and osteoporotic patients, with increased Firmicutes phyla and reduced Bacteroidetes. This is explained, among other things, because the microbiome influences oxidative reactions and glycolysis. A healthy microbiome maintains the natural barriers of the endothelium with its mucus shield. Whereas dysbiosis, by destroying the barrier, can lead to increased permeability and the passage of endotoxins. Increased numbers of Coliform bacteria and Ralstonia genus combined with dysbiosis are associated with multiple sclerosis, Parkinson's disease, amyotrophic lateral sclerosis, and Alzheimer's disease. Furthermore, a dysbiosis-immune system/central nervous system (CNS) crosstalk is discussed. The vagus nerve between the CNS and the enteric nervous system is seen as a neuroendocrine link [49]. It also appears that it is necessary to study the entire microbiota and not only the bacterial microorganisms. Several studies have already shown that fungal and viral infections are associated with the increased abundance of bacteria [50,51]. In patients infected with influenza, an increased number of Pseudomonadales (not Pseudomonas aeruginosa) could also be detected in the upper respiratory tract. Further studies must show whether these have a pathological value. In the course of a viral infection and with its healing, the altered microbiome changes again in the direction of healthy individuals [51]. The extent to which viruses, fungi, and bacteria interact in critically ill patients and whether microorganisms found have a pathophysiological effect or are part of physiological colonisation must be shown in further investigations that examine, among other things, the molecular interactions between the microorganisms themselves.

Also, the examination of the microbiome alone may not be sufficient. A healthy intestinal microbiome seems to protect against Clostridioides difficileinfection, but there may also be other factors. For example, a metabolome study showed that the secondary bile acid deoxycholate acted as an inhibitor for Clostridioides difficile. Thus, secondary bile acid also appears to be a factor in Clostridioides difficile infection [52]. This study is another indicator that our microbiome interacts directly with our body and the goal must be to keep this interaction in balance.

To keep the gut microbiome healthy, McClave et al. recommend "the delivery of early enteral nutrition, the provision of soluble fibre, and the generation of short-chain fatty acids". As mentioned above, therapy with probiotics is discussed to achieve a healthy microbiome. It has been shown in vitro that Lactobacillus plantarum, Lactobacillus rhamnosus GG, and Lactobacillus casei have an anti-inflammatory effect on the epithelial tissue [49]. However, a large multicentre RCT could not demonstrate any impact between administering a probiotic with Lactobacillus rhamnosus GGcompared to placebo on the incidence of ventilator-associated pneumonia and other infections such as bacteremia [53]. Therefore, it needs to be further investigated whether sepsis patients can also benefit from probiotics. Alternatively, faecal microbiome transplants should be considered for critically ill patients [54]. It was possible to successfully treat sepsis with diarrhoea and multi-organ failure in critically ill patients by faecal microbiota transplant (FMT) [55-57]. Simultaneously, there are also reports of drug-resistant E. coli infection transmitted through a faecal microbiome transplant and death in due course [58]. Therefore, FMT should not be undertaken lightly, and donors must be very carefully selected and screened. It may be possible to identify the composition of a healthy microbiome to initiate targeted therapies in the future.

Many studies have focused on the microbiome in sepsis or sepsis-related complications consisting of either animal models, case series, or relatively small controlled studies. Therefore application to clinical practice is still difficult. Multicentre studies with greater power need to further verify the previous results and provide more detailed insights into the microbiome of the septic patient.

  1. Reviewer’s suggestion:

At last, please revise the test for language and grammar style, and to correct certain typing errors as, but not only, the following:

Don't use italics for "spp.", and use it only for microbial names (i.e., Candida)

Lines 237-238 (and subsequent); C, difficile is currently named Clostridioides difficile

Line 267:  Don't use italics for "et al."

and more

Authors’ statement:

We have revised the review concerning language and grammar style and have adjusted the nomenclature.

Related text sections:

Corrections were made throughout the manuscript.

Reviewer 3 Report

In this review paper, Tourelle and colleagues have outlined the current knowledge on analyses of the microbiome particularly in the perioperative sepsis patients and its correlation with patients’ pathologic course. This manuscript was well written despite it is a bit descriptive. Inclusion of any illustration(s) (e.g., pie chart) showing the tendency of sepsis-associated compositions of microbiome using the authors’ current accumulated data, if available, would be even further helpful for general readership. I have additional brief comments below, which can be promptly addressed by the authors at convenience.

Authors made a statement that utilization of next-generation sequencing (NGS) would be instrumental to diagnose and predict severity of sepsis patients without any description on the samples containing microbiome (feces, BAL, or blood?), especially in the Abstract.

Regarding Figure 1:

  • The citation of this Figure doesn’t seem to be in the text. It needs to cite this Figure with additional explanations.
  • Is marking numbers (1 through 7) needed? It seems unnecessary to me.
  • Font size of the descriptions is small.
  • The descriptions on organ-specific pathologic features look somewhat disorganized, which may convey authors’ message incompletely. Because there are no additional explanations in the figure legends, the descriptions need to be revised to be more concise. 

Author Response

Dear Reviewer,

Please find enclosed the revised version of the manuscript entitled “Sepsis and the human microbiome. Just another kind of organ failure? A review.” and a detailed point-to-point response to the reviewers’ comments.

We hope you will find the revised version appropriate for publication in Journal of Clinical Medicine.

Yours sincerely,

Felix Schmitt, MD

Response to the suggestions of Reviewer 3:

  1. Reviewer’s comment:

In this review paper, Tourelle and colleagues have outlined the current knowledge on analyses of the microbiome particularly in the perioperative sepsis patients and its correlation with patients’ pathologic course. This manuscript was well written despite it is a bit descriptive. Inclusion of any illustration(s) (e.g., pie chart) showing the tendency of sepsis-associated compositions of microbiome using the authors’ current accumulated data, if available, would be even further helpful for general readership. I have additional brief comments below, which can be promptly addressed by the authors at convenience.

Authors’ statement:

We would like the thank the reviewer for the constructive and helpful comments. We already discussed the possibility to summarize the sepsis-associated changes. But we have not found a good solution to display the changes in the composition, because of the heterogeneity of the data. But we overworked figure 1 for a better readability and more clarity.

  1. Reviewer’s comment:

Authors made a statement that utilization of next-generation sequencing (NGS) would be instrumental to diagnose and predict severity of sepsis patients without any description on the samples containing microbiome (feces, BAL, or blood?), especially in the Abstract.

Authors’ statement:

To further improve our abstract, we specified the most commonly used samples in the studies.

Related text sections:

Abstract: Next-generation sequencing (NGS) has been further optimised during the last years and has given us new insights into the human microbiome. Especially the 16S-rDNA sequencing is a cheap, fast, and reliable method that can reveal significantly more microorganisms compared to culture-based diagnostics. It might be a useful method for patients suffering from severe sepsis and at risk of organ failure because early detection and differentiation between healthy and harmful microorganisms are essential for effective therapy. In particular, the gut and lung microbiome have been examined with NGS in critically ill patients. For this review, an iterative approach was used. Current data suggest that an altered microbiome with a decreased alpha-diversity compared to healthy individuals could negatively influence the individual patient's outcome. In the future, NGS may not only contribute to the diagnosis of complications. Patients at risk could also be identified before surgery or even during their stay in an intensive care unit. Unfortunately, there is still a lack of knowledge to make precise statements about what constitutes a healthy microbiome, which patients exactly have an increased perioperative risk, and what could be a possible therapy to strengthen the microbiome. This work is an iterative review that presents the current state of knowledge in this field.

  1. Reviewer’s comment:

Regarding Figure 1:

  • The citation of this Figure doesn’t seem to be in the text. It needs to cite this Figure with additional explanations.
  • Is marking numbers (1 through 7) needed? It seems unnecessary to me.
  • Font size of the descriptions is small.
  • The descriptions on organ-specific pathologic features look somewhat disorganized, which may convey authors’ message incompletely. Because there are no additional explanations in the figure legends, the descriptions need to be revised to be more concise. 

Authors’ statement:

We have now reorganized the figure, removed the numbering, adjusted the font size and expanded the description. Futhermore, we have cited the graphic in the text.

Related text sections:

In a mouse model, it was shown that a high-fibre rodent diet could ensure that the inflammatory response to sepsis was lower than in mice without a special diet [27]. A pilot study showed no significant differences in beneficial short-chain fatty acids in critically ill patients on broad-spectrum antibiotic therapy who received either a regular diet or a high-fibre diet. There was just a trend towards a high-fibre diet and increased beneficial short-chain fatty acids, but the number of cases was small (10 patients per group) [28]. This reflects that altering the gut microbiota can have an impact on sepsis and its complications. An individualised diet adapted to the patients, and their microbiome, their disease, their pre-existing conditions, and their general condition should be the goal to avoid the loss of alpha diversity and an accompanying dysbiosis [34]. Treatment with probiotics containing L. plantarum reduces infections in critically ill patients [35]. A combination of probiotics and prebiotics, which include non-digestible fiber and can promote the growth of certain bacteria, are called synbiotics. The administration of synbiotics in critically ill patients increased beneficial bacteria such as Bifidobacterium and Lactobacillus in the stool. At the same time, there were significantly fewer cases of enteritis and VAP in the group receiving synbiotics. However, an advantage in terms of mortality could not be shown [29].

An overview of the most important key findings of perioperative sepsis, sepsis-induced ARDS, and the influence of drugs and probiotics is shown in Figure 1.

[Graphic 1]

Figure 1. Overview of the key findings. There are many ways to influence the microbiome in critically ill patients with sepsis and ARDS. For example, antibiotics and opioids harm the balance of the microbiome. On the other hand, it appears that probiotics have a positive influence on dysbiosis. Furthermore, typical gastrointestinal bacteria can be detected in the lungs of ARDS patients and are indicative of lung dysbiosis. Overall, reduced alpha diversity in the gut and lungs is associated with more complications and increased hospital mortality. ARDS – acute respiratory distress syndrome, NHAI-pneumonia - nursing-home and hospital-associated infection-pneumonia

Reviewer 4 Report

Title “Sepsis and the human microbiome. Just another kind of organ failure? A review.” Is non-specific and the word ‘organ failure’ appears only 6 times. Do you mean “Sepsis and the human microbiome. A review of the knowledge gaps.”

The abstract is vague and short on specifics. For example, what is “…a hampered microbiome ..”? I presume that 21 studies [ref 9-29] are reviewed – correct?

The manuscript is poorly written for example

Line 47 “..coincided with the blood cultures,..” do you mean “..coincided with the results of blood cultures,..:

Line 48 “..In comparison, classical blood cultures..” do you mean “..In comparison, conventional blood culture methods”

Line 50 “..result from decontamination..” is completely unclear.

Line 56 “…increased positive rate in the course of sepsis” is unclear – positive for what?

Figure 1 is potentially misleading without specific study citations to support the figure.

The objectives of the manuscript [lines 69-74] could be sharpened to a specific patient group, end point and possibly even intervention [PICOS]. Likewise the first para of the materials and methods.

Normally I would expect to see a description of how the synthesis of the information was done, how many studies were found [this number needs to be in the abstract], how many were excluded for being inadequate and for what reason, etc, etc. At the very least, you need to state the studies were appear to have been selected to fill out Table 1.

The conclusion also mentions ‘organ failure’ – why? The conclusion also mentions ‘peri-operative setting’ – why?

Author Response

Dear Reviewer,

Please find enclosed the revised version of the manuscript entitled “Sepsis and the human microbiome. Just another kind of organ failure? A review.” and a detailed point-to-point response to the reviewers’ comments.

We hope you will find the revised version appropriate for publication in Journal of Clinical Medicine.

Yours sincerely,

Felix Schmitt, MD

Response to the suggestions of Reviewer 4:

  1. Reviewer’s suggestion:

Title “Sepsis and the human microbiome. Just another kind of organ failure? A review.” Is non-specific and the word ‘organ failure’ appears only 6 times. Do you mean “Sepsis and the human microbiome. A review of the knowledge gaps.”

Authors’ statement:

To better reflect our title, we have integrated more the term “organ failure” into our text.

Related text section:

Additions were made throughout the manuscript.

  1. Reviewer’s suggestion:

The abstract is vague and short on specifics. For example, what is “…a hampered microbiome ..”? I presume that 21 studies [ref 9-29] are reviewed – correct?

Authors’ statement:

According to the Reviewers suggestion, we have rewritten the abstract to provide a more appropriate overview of the underlying review.

Related text sections:

Abstract: Next-generation sequencing (NGS) has been further optimised during the last years and has given us new insights into the human microbiome. Especially the 16S-rDNA sequencing is a cheap, fast, and reliable method that can reveal significantly more microorganisms compared to culture-based diagnostics. It might be a useful method for patients suffering from severe sepsis and at risk of organ failure because early detection and differentiation between healthy and harmful microorganisms are essential for effective therapy. In particular, the gut and lung microbiome have been examined with NGS in critically ill patients. For this review, an iterative approach was used. Current data suggest that an altered microbiome with a decreased alpha-diversity compared to healthy individuals could negatively influence the individual patient's outcome. In the future, NGS may not only contribute to the diagnosis of complications. Patients at risk could also be identified before surgery or even during their stay in an intensive care unit. Unfortunately, there is still a lack of knowledge to make precise statements about what constitutes a healthy microbiome, which patients exactly have an increased perioperative risk, and what could be a possible therapy to strengthen the microbiome. This work is an iterative review that presents the current state of knowledge in this field.

  1. Reviewer’s suggestion:

The manuscript is poorly written for example

Line 47 “..coincided with the blood cultures,..” do you mean “..coincided with the results of blood cultures,..:

Line 48 “..In comparison, classical blood cultures..” do you mean “..In comparison, conventional blood culture methods”

Line 50 “..result from decontamination..” is completely unclear.

Line 56 “…increased positive rate in the course of sepsis” is unclear – positive for what?

Authors’ statement:

We want to apologize that our writing styles have not pleased the reviewer. We changed the examples the reviewer mentioned. 

Related text sections:

In 2010, a study established a catalogue of 3.3 million non-redundant human intestinal microbial genes [1]. NGS is a novel procedure among molecular methods for pathogen diagnostics, and it is able to analyse the overall DNA fragments in the sample. The procedure can also differentiate between human DNA, bacteria, eukaryotes, archaea, and chloroplasts. In 2013, Skvarc et al. investigated molecular methods for pathogen diagnostics using PCR. At that time, the conclusion was that, despite certain limitations, molecular methods should be further investigated together with culture-based methods in everyday clinical practice [2]. Since the NGS method has broader applications than previous molecular methods, this statement is more valid than ever. NGS was also able to determine which bacteria were present in the blood of critically ill patients within 30 hours. The results coincided with the results of blood cultures, and NGS also showed other microbiota that were not detectable in blood cultures [3]. In comparison,conventional blood cultures take between 24 and 120 hours from collection to results [4]. Cell-free DNA found in NGS may also result from the translocation of DNA by gut bacteria into the blood. Therefore, NGS results from blood samples should be interpreted critically. Initial work on cell-free DNA sequencing with optimised workflows and nanopore-based real-time sequencing, which further develops NGS, detected bacteremia within 6 hours. It was also shown that this method achieved a detection rate of 90.6% in 239 retrospectively evaluated sepsis samples by extrapolation. Furthermore, by using NGS, the study investigated a 6-fold increased positive rate for pathogen hits in the course of sepsis compared to blood cultures [5]. In another study with septic patients, NGS results would have resulted in a change in antibiotic therapy in 53% of cases [6]. Also, in pneumonia patients in an ICU, NGS could detect microbiota in 84% of the patients, whereas culture based methods could only detect microbiota in 65% of the patients [7]. Otto et al. were able to show that there are three phases during sepsis and that these are probably related to changing pathophysiological mechanisms. In the late phase of sepsis, there is an increase in positive blood cultures [8]. In combination with the previously presented results, showing that blood cultures only select some of the microbiota, NGS methods could provide earlier and further insights into the pathophysiological mechanisms of sepsis and a possible organ failure. However, it must be emphasized that NGS results must then be examined to determine whether the pathogens found were viable or dead pathogen residues. For this purpose, the wet lab or bioinformatics must reconcile the results critically.

  1. Reviewer’s suggestion:

Figure 1 is potentially misleading without specific study citations to support the figure.

Authors’ statement:

We appreciate the reviewer's feedback. There is no citation because the figure is a graphic summary of the findings in our results.

  1. Reviewer’s suggestion:

The objectives of the manuscript [lines 69-74] could be sharpened to a specific patient group, end point and possibly even intervention [PICOS]. Likewise the first para of the materials and methods.

Authors’ statement:

We appreciate the reviewer's feedback. Since this Review works as an overview for the microbiome of critically ill patients with sepsis or ARDS, we used an iterative method. We had no specific endpoint nor intervention, nor comparison group we could integrate into this review.

  1. Reviewer’s suggestion:

Normally I would expect to see a description of how the synthesis of the information was done, how many studies were found [this number needs to be in the abstract], how many were excluded for being inadequate and for what reason, etc, etc. At the very least, you need to state the studies were appear to have been selected to fill out Table 1.

Authors’ statement:

As mentioned in Material & Methods, we chose an iterative approach for this review and condensed papers by information content. After an initial overview, further research was conducted based on the data seen so far. Therefore, a fixed number of papers were not found in advance, like in a systematic review. It is therefore difficult to quantify the number of papers screened with certainty. To make the selected papers as transparent as possible, the table with the essential points was created.

  1. Reviewer’s suggestion:

The conclusion also mentions ‘organ failure’ – why? The conclusion also mentions ‘peri-operative setting’ – why?

Authors’ statement:

We have chosen the keywords "microbiome", "microbiota", "perioperative complications", "sepsis", "pneumonia", "ARDS" or "16S RNA" for this review. The corresponding papers are listed in the results and integrated into the discussion, and therefore we have chosen the terminology.

Round 2

Reviewer 3 Report

I have minor comments to the revised manuscript. The authors should address them.

- Line 22: The sentence “In particular, the gut and lung microbiome have been examined with NGS in critically ill patients” needs to be revised.

My suggestion: In particular, the gut and lung microbiome in critically ill patients have been probed by NGS.

- Line 289: Or if we may detecting…. ??? This sentence needs to be fixed.

- Line 303: endothelium?? Is it correct? Please double-check if it is “endothelium” or “epithelium”. And include additional reference(s) in the paragraph (lines 297~306) if the contents were referred to.

Author Response

Dear Reviewer,

Please find enclosed the revised version of the manuscript entitled “Sepsis and the human microbiome. Just another kind of organ failure? A review.” and a detailed point-to-point response to the reviewers’ comments.

We hope you will find the revised version appropriate for publication in Journal of Clinical Medicine.

Yours sincerely,

Felix Schmitt, MD

Response to the Reviewer

Response to the suggestions of Reviewer 3:

  1. Reviewer’s suggestion:

I have minor comments to the revised manuscript. The authors should address them.

- Line 22: The sentence “In particular, the gut and lung microbiome have been examined with NGS in critically ill patients” needs to be revised.

My suggestion: In particular, the gut and lung microbiome in critically ill patients have been probed by NGS.

- Line 289: Or if we may detecting…. ??? This sentence needs to be fixed.

- Line 303: endothelium?? Is it correct? Please double-check if it is “endothelium” or “epithelium”. 

And include additional reference(s) in the paragraph (lines 297~306) if the contents were referred to.

Authors’ statement:

We are grateful for the valuable suggestions of the reviewer. We have revised the review and incorporated the recommendations. Unfortunately, the line references were not congruent with those in the current manuscript, but we have identified and improved the text passages.

Related text section:

Line 22:

… It might be a useful method for patients suffering from severe sepsis and at risk of organ failure because early detection and differentiation between healthy and harmful microorganisms are essential for effective therapy.  In particular, the gut and lung microbiome in critically ill patients have been probed by NGS. For this review, an iterative approach was used. Current data suggest that an altered microbiome with a decreased alpha-diversity compared to healthy individuals could negatively influence the individual patient's outcome. …

Line 372 (289):

Whether and to what extent these drugs have an impact on the microbiome and consequently on the outcome during sepsis remains unclear. It also needs to be clarified whether, for example, the reduced gut motility due to opioids and not the opioids themselves are responsible for a lowered alpha diversity.

Line 382-393 (303 & 297-306):

Oral gut dysbiosis is associated with metabolic and degenerative diseases [49]. These include metabolic syndrome, type 2 diabetes and chronic neurodegenerative diseases [50,51]. In addition, intestinal dysbiosis has been shown in osteopenic and osteoporotic patients, with increased Firmicutes phyla and reduced Bacteroidetes [52]. This is explained, among other things, by changes inoxidative reactions and glycolysis [53,54]. A healthy microbiome maintains the natural barrier of the epithelium with its mucus shield. Whereas dysbiosis, by destroying the barrier, can lead to an increased permeability and the migration of endotoxins. Dysbiosis is associated with neurodegenerative disorders like Alzheimer's and Parkinsons disease [55,56]. Furthermore, a dysbiosis-immune system/central nervous system (CNS) crosstalk is discussed. The vagus nerve between the CNS and the enteric nervous system is seen as a neuroendocrine link [51,56].
